# Poyang Lake Wetland Classification Using Time-Series ENVISAT ASAR Data and Beijing-1 Imagery

**Fang Ding** [1,2,3], **Lin Wang** [2,3,4,*], **Iryna Dronova** [5] and **Kun Cao** [2,3]

1　College of Marine Sciences, Shanghai Ocean University, Shanghai 201306, China
2　Fishery Resource and Environment Research Center, Chinese Academy of Fishery Sciences, Beijing 100141, China
3　Scientific Observing and Experimental Station of Fishery Remote Sensing, Ministry of Agriculture and Rural Affairs, Beijing 100141, China
4　State Key Laboratory of Remote Sensing Science, Jointly Sponsored by the Institute of Remote Sensing Applications of Chinese Academy of Sciences and Beijing Normal University, Beijing 100101, China
5　Department of Environmental Science, Policy, and Management, Rausser College of Natural Resources, University of California Berkeley, Berkeley, CA 94720, USA
*　Correspondence: angels121@cafs.ac.cn; Tel.: +86-010-6867-3951

**Abstract:** Beijing-1 and ENVISAT ASAR images were used to classify wetland aquatic macrophytes in terms of their plant functional types (PFTs) over the Poyang Lake region, China. Speckle noise filtering, systematic sensor calibration within the same polarization or between different polarizations, and accurate geo-registration were applied to the time-series SAR data. As a result, time-series backscattering data, which is described as permittivity curves in this paper, were obtained. In addition, time-series indices, described as phenological curves, were derived from Beijing-1 time-series images in the classification experiment. Based on these two curves, a rule-based classification strategy was developed to extract wetland information from the combined SAR and optical data. In the rule-based wetland classification method, DEM data, submersion time index, temporal Beijing-1 images, time-series normalized difference vegetation index (TSNDVI) images, principal component analysis (PCA), and temporal ratio of ASAR time-series images were used. In addition, a decision tree-based method was used to map the wetlands. Conclusions include the following: (1) after the preprocessing of ASAR data, it was possible to satisfactorily separate different aquatic plant functional types; (2) hydrophytes from different PFTs exhibited distinct phenological, structural, moisture, and roughness characteristics due to the impact of the annual inundation of Poyang Lake wetland; and (3) more accurate results were obtained with the rule-based method than the decision tree (DT) method. Producer's and user's accuracy calculated from test samples in the classification results indicate that the DT method can potentially be used for mapping aquatic PFTs, with overall producer's accuracy exceeding 80% and higher user's accuracy for aquatic bed wetland PFTs. A comparison of producer's and user's accuracy from the rule-based classification increased from 3 to 12% and 7 to 26%, respectively, for different aquatic PFTs.

**Keywords:** plant functional type; time-series remote sensing images; phenological curves; permittivity curves; rule-based classification method

## 1. Introduction

Wetland ecosystems worldwide have undergone significant losses in area and biological diversity [1–5]. Furthermore, due to the difficulty accessing most sites, the lack of detailed knowledge on the distribution of wetland vegetation has become a significant barrier to understanding and managing wetlands.

Wetland mapping and protection in China has become more of a concern worldwide [2,6–9]. Poyang Lake is the largest freshwater lake in China. It is one of the 10 most important ecological functional conservation areas in the country [10], and one of the largest

bird conservation areas and habitats for migratory birds in the world because of its suitable environment and climatic conditions in winter [11]. More than 98% of the total population of critically endangered Siberian crane (*Grus leucogeranus*) use the lake area as their winter habitat. It is also serves as a crucial habitat for cetaceans and various fish species. Poyang Lake is known as the main shelter of the Yangtze finless porpoise, as it holds approximately half of the total population. The high connectivity of the channel connecting Poyang Lake and the Yangtze River not only ensures normal migration of the four major Chinese carp in the middle reaches of the river, but also makes the lake a feeding and fattening field for them [12].

Due to the annual flood cycle and significant water level fluctuation of Poyang Lake, the distribution of aquatic vegetation is strongly affected by water level fluctuation and hydrological conditions. The boundaries of different aquatic communities have shown a directional shift in recent decades, from emergent vegetation to sedges and forbs and submerged vegetation [13]. Furthermore, the annual average water level and surface water occurrence frequency have been decreasing in recent years. The shift of vegetation boundaries may be an indicator of natural or human-induced environmental changes [14,15], which in turn can have a considerable impact on ecosystem functions, economic development, and carrying capacity of wetlands, as well as ecological services for humans and other species in the Poyang Lake region.

For all of these reasons, a plan is being proposed to construct a floodgate at the outlet of Poyang Lake, with the aim of regulating the lowest water level to ensure an adequate amount of water in winter. However, the potential impacts of this measure on the abundance and distribution of wintering waterbirds [16,17], Yangtze finless porpoises [18,19], and other wildlife and the function of the entire Poyang Lake wetland ecosystem are still unknown due to the lack of information on aquatic vegetation and accurate bathymetry data of the lake bottom in this region. Therefore, there is an urgent need to map the spatial distribution of aquatic vegetation at the plant functional level for the purpose of environmental simulation and wildlife habitat modeling under scenarios of future climate change and anthropogenic alterations of the landscape. The concept of plant functional type (PFT) has received an increasing amount of attention due to the need for generalization in biogeographic and ecological modeling [20–24] for improved understanding of environmental and global changes. However, there are relatively few studies on mapping PFTs in terms of remote sensing [20,25–30], especially with regard to natural wetland ecosystems at the regional scale.

In systems with strong seasonal differences in vegetation structure and appearance, multi-temporal imaging can be particularly useful for PFT-level discrimination. Optical remote sensing for mapping aquatic macrophytes in water bodies has a number of limitations. Submerged plants are separated from the optical sensor by a water layer, which also reflects and absorbs incoming and outgoing radiation and may contain sediment and particles that obscure live plant biomass and aquatic canopy architecture from the sensor. Variability in water depth across the study area can also influence the interpretability of data from optical remote sensing and the capability to detect the spatial distribution and spectral characteristics of aquatic macrophyte beds. As the height of the water column increases, there is attenuation of light reaching aquatic vegetation, which constrains plant photosynthesis and results in zonation of plant distribution from the shore to deeper areas. Finally, atmospheric clouds and haze can block the view from an optical sensor, especially if the water bodies of interest are typically covered by clouds during the season when the state of vegetation is of particular interest ecologically and phenologically. Synthetic aperture radar (SAR), with its unique ability to detect flooding beneath vegetation canopies and penetrate cloud cover, can provide more frequent information than optical remote sensing for water extent monitoring and aquatic macrophyte analysis [31–35]. However, the speckle effect on all SAR images clearly constrains the analysis of SAR data. Therefore, it is desirable to combine the strengths of optical and SAR data in wetland monitoring using a multi-sensor and multi-temporal approach.

The objectives of this study were as follows: (1) to analyze time-series spectral characteristics and backscattering signatures of different aquatic plant functional types (PFTs), and (2) to test and improve a new rule-based classification method for distinguishing aquatic PFTs in the Poyang Lake wetlands.

## 2. Materials

### 2.1. Study Area

Poyang Lake is located between 28°22′–29°45′ N and 115°47′–116°45′ E. It lies in the northern part of Jiangxi Province, China, at the southern bank of the lower reach of the Yangtze River (Figure 1). The basin has an average annual temperature of 17 °C, annual rainfall of 1400–1900 mm, and a 240–330 day frost–free period [36,37]. It is 173 km long from north to south, and the mean width is 16.9 km from west to east, with the largest width being 74 km. The lakeshore is 1200 km long and covers a water area of 3569 km² (when the water level at Hukou station is 17.9 m). Five rivers (Gan, Fu, Xiu, Rao, and Xin Rivers) drain through Poyang Lake into the Yangtze River at Hukou. Most of the annual precipitation falls between April and Jun, and during this period the lake fills up, with the water from the five rivers covering all of its bottomland and some marshlands. From July to September, water levels reach a peak with backflow from the Yangtze River. In October and November, the water surface subsides and vast tracts of flat grass-covered marshlands emerge above the water. The water body area varies greatly with fluctuations in its level throughout the year. This forms a special scenery of marshlands in winter and floods in summer [30,38,39] over the lake area.

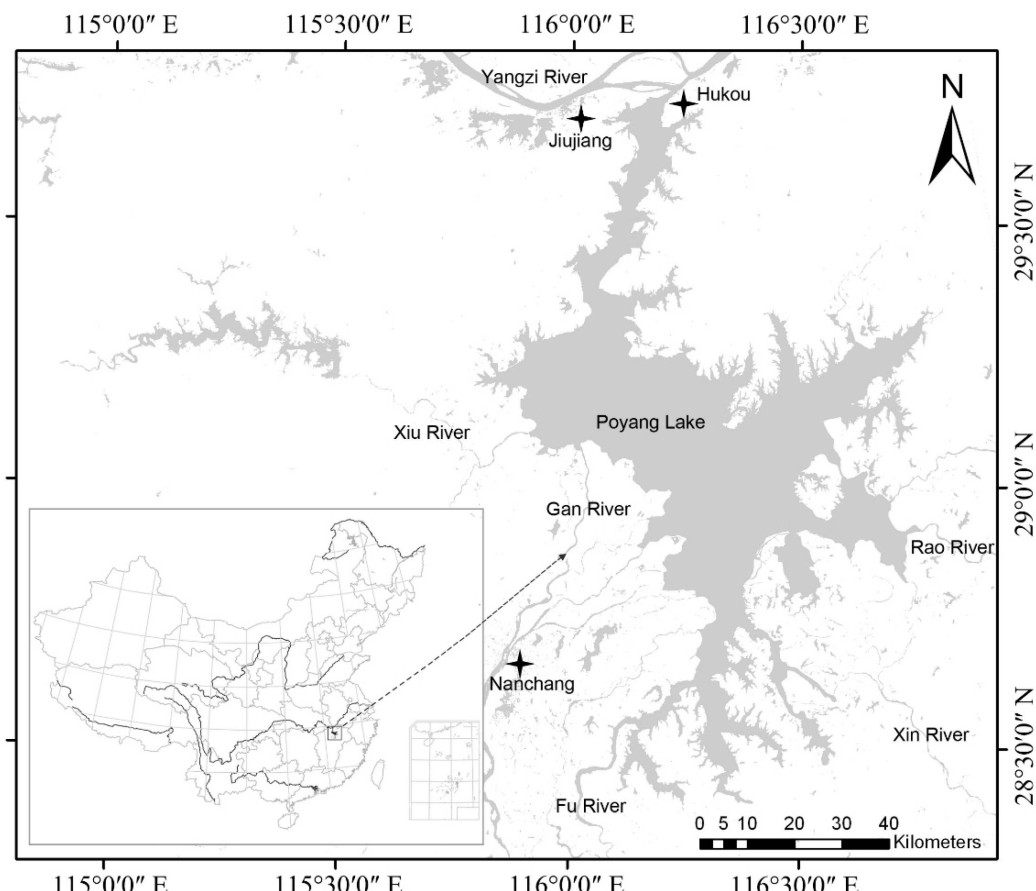

**Figure 1.** Research area in this study.

### 2.2. Image Data and Preprocessing

The Beijing-1 small satellite was launched on 27 October 2005 with one 4 m panchromatic and two 32 m multi-spectral sensors on board. Each multi-spectral sensor has 3 bands, covering green (0.52–0.62 μm), red (0.63–0.69 μm), and near-infrared (0.76–0.9 μm), with 8 bit quantization and a 320 km swath. We used 10 scenes from Beijing-1 multi-spectral images, each acquired on a different date from April 2007 to May 2008 (Table 1).

**Table 1.** Satellite data used in this study.

| Satellite | Year | Dates | Water Level (m) [a] | GCPs [b] | Clouds Over Lake | RMSEs [c] |
|---|---|---|---|---|---|---|
| Beijing-1 | 2007 | 19 April, 6 May, 26 July, 16 August, 17 October, 30 November | 10.78, 13.04, 17.02, 17.53, 11.94, 8.71 | 21, 315, 28, 26, 25, 24 | 0, <2%/NO, 0, <2%/NO, <2%/YES, 0 | 0.48, 0.93, 0.43, 0.49, 0.50, 0.41 |
| | 2008 | 1 January, 16 February, 2 March, 12 May (10 scenes) | 8.59, 9.61, 8.89, 12.75 | 28, 28, 25, 32 | 0, 0, 0, <2%/YES | 0.89, 0.45, 0.34, 0.45 |
| ASAR WSM | 2007 | 25 May, 10 June, 29 June, 31 July (VV), 4 September, 28 October (VV), 18 December | 10.22, 12.84, 15.98, 17.57, 16.16, 9.95, 8.20 | 23, 22, 19, 20, 18, 30, 27 | Free of clouds | 0.36, 0.32, 0.33, 0.31, 0.43, 0.45, 0.49 |
| | 2008 | 3 Jan, 7 February, 1 April, 20 April (11 scenes) | 8.51, 10.73, 11.62, 13.55 | 46, 30, 33, 25 | | 0.43, 0.38, 0.49, 0.36 |

Note: We used 10 scenes of Beijing-1 multispectral images with 32 m spatial resolution, and 11 scenes of ASAR WSM images with 150 m spatial resolution. [a] Water level data correspond to scene acquisition date, measured at Duchang Hydrological Station. [b] Number of ground control points used for georegistration. [c] Root mean square error from georegistration for each scene. Landsat TM data (30 m resolution) acquired on 10 December 1999 were used as a base on which other Beijing-1 images were registered.

The ENVISAT satellite was launched by the European Space Agency on 1 March 2002. The advanced synthetic aperture radar (ASAR) instrument operates at C-band (5.3 GHz) and has several polarizations: HH, VV, HV, and VH. The incidence angles and spatial/radiometric resolution of ASAR depend on the functioning mode (ENVISAT ASAR Product Handbook, 2004). The ASAR instrument has two operation modes, conventional stripmap SAR (image and wave modes) and ScanSAR (global monitoring, wide swath, and alternating polarization modes) [40–42]. We used 9 scenes of ASAR images in wide swath mode (WSM) at HH polarization and 2 scenes at VV polarization (Table 1). The spatial resolution is 150 m and the incidence angle is 43° (ENVISAT Handbook, 2004).

The ASAR image intensity was converted into backscattering coefficient σ° using the Basic ENVISAT SAR Toolbox (BEST) software provided by ESA. Speckle noise in the image of backscattering coefficient σ° was suppressed by using the Lee adaptive despeckle filter [43,44] with a 9-by-9 window size. Then, we performed two steps to calibrate the ASAR data within the same polarization data and between different polarization data. Similarly, all Beijing-1 optical images were converted to apparent surface reflectance to eliminate radiometric differences and remove solar angle differences. Details on the calibration of ASAR data and Beijing-1 images are given by Lin et al. [45].Next, the calibrated backscattering coefficient σ° images and Beijing-1 images were geo-referenced to each other. The geo-referencing was performed using ENVI 5.1 (Exelis Visual Information Solutions, Boulder, CO, USA), and the results were assessed by superimposing Beijing-1 and ASAR images onto a Landsat TM image (30 m resolution) acquired on 10 December 1999, and the geometric resolution of both resulting images was resampled to 32 m. We did not conduct topographic correction, because the research area is flat. Some characteristics of the images are summarized in Table 1. The Beijing-1 image acquired on 5 June 2007 produced abnormal geometrical distortion, so we used considerably more ground control points to make sure that the resultant RMSE was within a pixel edge length.

### 2.3. Hydrological Data Collection and Analysis

We acquired daily water level data measured at Duchang Hydrological Station between 1 January 2007 and 31 December 2008. Among the 7 hydrological stations around

Poyang Lake, the water level of the lake measured by this station best correlates to the water area of the lake [46].

In order to understand the inter-annual water dynamics of Poyang Lake for further analysis, we calculated the statistical properties of water level data, including monthly mean, maxima, and minima (Figure 2), from 1 January 2007 to 31 December 2008. From these results, we found that the change patterns of monthly mean, maxima, and minima were consistent in the 2 years. Temporal change patterns were relatively stable and comparable between the 2 years, especially during periods of water infill, high water levels, and subsiding water, implying that Poyang Lake should have experienced similar vegetation change patterns according to the water level changes during the 2-year period. These findings suggest that no extreme weather events occurred during those 2 years.

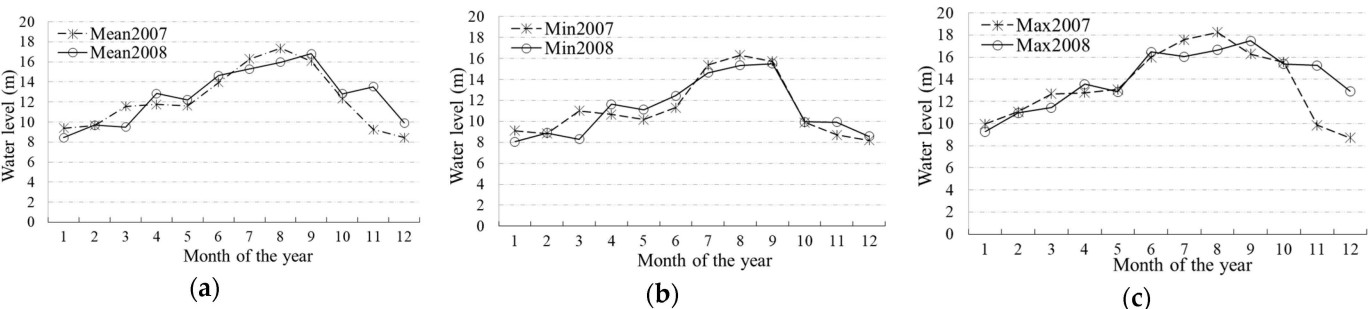

**Figure 2.** Monthly (**a**) mean, (**b**) minima, and (**c**) maxima of water level (m) between 2007 and 2008.

## 3. Methods

### 3.1. Classification System and the Field Sampling

We propose a set of PFT classifications (Figure 3) based on the field surveys and referring to existing wetland classification systems [3,47,48]. The field surveys were conducted in April 2006, September 2006, March 2007, June 2007, July 2007, December 2007 to January 2008, March 2008, April to May 2008, November 2008, and May 2009 (Figure 4). We mainly considered vegetation traits: photosynthesis pathway, morphology, dominant species composition, structural features, growth, and phenological characteristics (Table 2). Our PFT classification scheme includes the following subclasses: emergent aquatic macrophytes, giant C4 grasses, C3 (sedges and forbs), and floating and submerged aquatic macrophytes. We also considered other land cover types, such as sand, mud, and seasonal water, in the analysis. Although the latter are not aquatic plant functional types, they are useful in scaling up the model and monitoring the distribution of different PFTs during the annual inundation period. The definitions and descriptions of the system, subsystem, and classes in this paper are those given by Cowardin et al. [47]. The five primary PFT subclasses, shown in Table 2, exhibit zonal distribution along the environmental gradients of water table depth and inundation duration [13,49] from the lake shore to the lake center.

Site-specific information on plant species and traits related to characteristics of different PFTs (Table 2) was used to develop training samples for image classification. Photographs taken in the field also helped in training and validation sample selection. Some sample points were revisited several times to verify that they contained the same dominant PFTs during the growing seasons in the entire study period (Figure 4). From all the field data collected, we selected 237 samples for training (29 for giant C4 grasses, 28 for emergent aquatic macrophytes, 135 for C3 (sedges and forbs), 23 for floating aquatic macrophytes, and 22 for submerged aquatic macrophytes) and 305 samples for validation (30 for giant C4 grasses, 20 for emergent aquatic macrophytes, 220 for C3 (sedges and forbs), 16 for floating aquatic macrophytes, and 19 for submerged aquatic macrophytes). Their locations were recorded by GPS. The area of each field sample site was more than $60 \times 60$ m$^2$ (about $2 \times 2$ pixels on Beijing-1 images).

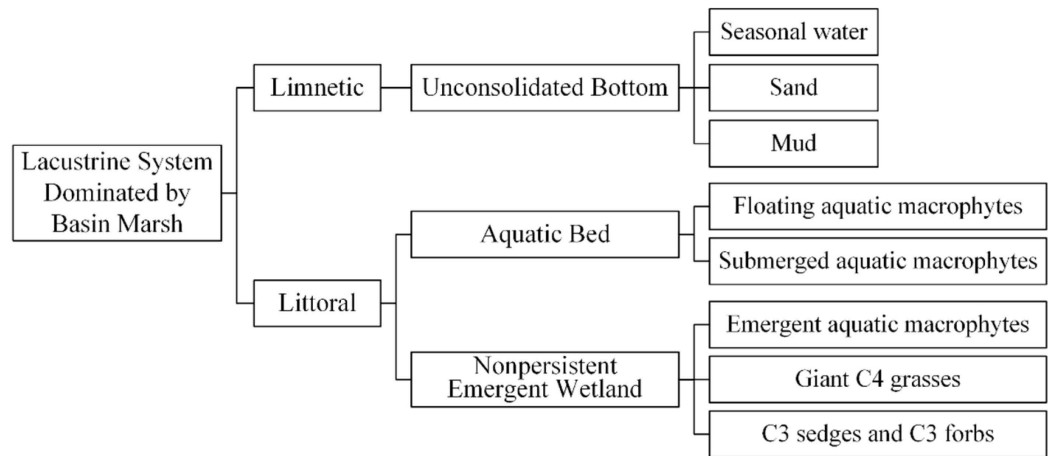

**Figure 3.** PFT classification scheme in this analysis.

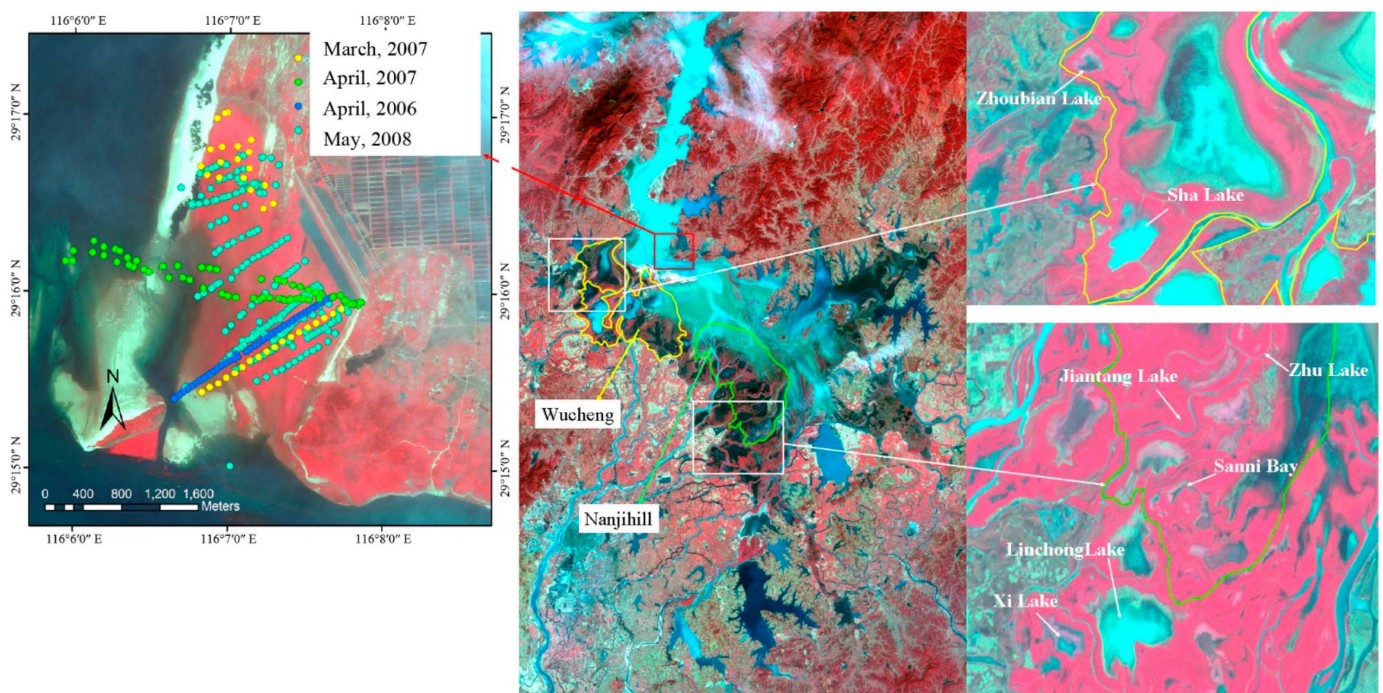

**Figure 4.** Example of sample validation locations from field visits in May 2009 and revisited field sample locations from 2006 to 2008.

**Table 2.** Plant functional type classification system for Poyang Lake.

| PFT Subclass | Dominant Species (>70–100%) | Relationship to Water Table Height | Mean Submersion Time/Months | Growing Season | Canopy Height above Soil (m) | Elevation of Distribution (Meters above Sea level) |
|---|---|---|---|---|---|---|
| Emergent aquatic macrophytes | *Phragmites communis, Zizania caduciflora, Phalaris arundinacea* | Water table from 0.5 m below soil surface to 1–2 m above soil | 1–2 months /July–August | April–October | 0.8–5 | 17.7–20 |
| Giant C4 grasses | *Miscanthus sacchariflorus, Arundinella hirta* | Upland dry, water table not greater than 1.7 m below soil surface to 1–2 m above soil | 2–3 months /July–September | April–October | 1–3 | 16–17.7 |
| C3 sedges and forbs | *Carex cinerascens, C. unisexualis,* sometimes mixed with *Artemisia* spp. | Water table from 0.5 m below soil surface to 1–3.5 m above soil | 3–5 months /May–October | April–May | 0.1–1.5 | 14.2–16 |
| Floating aquatic macrophytes | *Nymphoides peltata, Trapa bispinosa, Potamogeton malaianus* | Mud-forming floating on water surface at water height above soil surface at water depth 0.3–2 m | 5–7 months /April–November | September | 0.5–2 | 13.8–14.2 |
| Submerged aquatic macrophytes | *Potamogeton franchetii, Vallisneria spiralis L., Hydrilla verticillata, Ceratophyllum demersum* | Underwater | 7–12 months /January–December | December | 0.5–2.5 | <13.8 |

Note: In the classification, we also included mud, water, sand, farmland, and non-flooded forest land cover types.

### 3.2. Time-Series Data Analysis

We used time-series NDVI (TSNDVI) images of Beijing-1 (Table 1), which were expected to reflect the vegetation traits growth and phenology of dominant species. A TSNDVI image is a stacked file of 10 NDVI images. The corresponding image positions of training samples were used to extract the time-series curve from TSNDVI. For classification, we used 170 training samples for giant C4 grasses, 151 for emergent aquatic macrophytes, 334 for C3 (sedges and forbs), 81 for floating aquatic macrophytes, 101 for submerged aquatic macrophytes, 74 for mud, 165 for water, and 87 for sand; among them, field sample sites were composed of 29, 28, 135, 23, and 22 samples, respectively, for the five PFTs. Each field sample site included 2 to 10 image pixels, depending on the level of homogeneity and area surveyed. At each pixel location, NDVI values at different times were extracted from 10 Beijing-1 images to form a time-series curve, here referred to as phenological curve (Figure 5a).

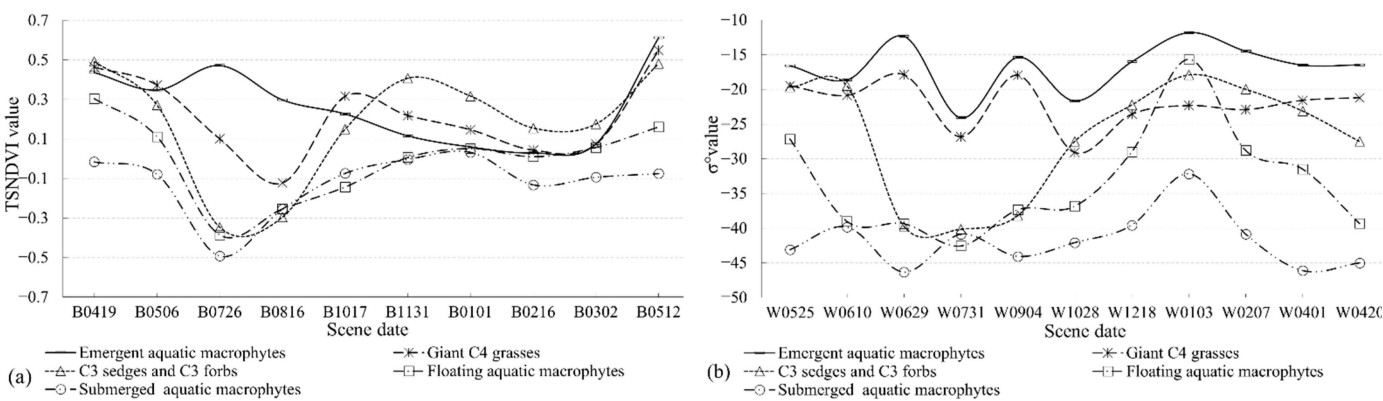

**Figure 5.** Time-series curves extracted from Beijing-1 and ASAR data: (**a**) phenological curve extracted from Beijing-1 time-series NDVI images; (**b**) permittivity curve extracted from ASAR time-series backscattering coefficients images. (Scene date is given as image source, month, day. B represents Beijing-1 and W represents ASAR WSM. For example, B0419 is Beijing-1 image acquired on 19 April).

The time-series ASAR WSM (TSWSM) images were stacked from the 11 ASAR WSM images (Table 1). This was expected to reflect the vegetation traits size, density, shape, and dielectric constant of the target, as well as SAR system characteristics such as incident angle, polarization, and wavelength [50–52]. At each training sample pixel location, backscattering coefficient σ° values at different times were extracted from the TSWSM to form a time-series curve, here referred to as permittivity curve (Figure 5b).

Based on the phenological and permittivity curves, we analyzed and discussed the mechanisms and reasons for their variations, and they were further used to build a rule-based classification model and decision tree model to perform freshwater wetland mapping.

### 3.3. Rule-Based and Decision Tree Approach to Time-Series Optical and Radar Data

Some approaches have been proposed that are aimed at constructing an accurate classifier, such as decision tree, rule-based system, and association-based classifiers. Among these, following a top-down and hierarchical paradigm, the decision-tree (DT) method, which directly constructs a classification model from the training data, has been demonstrated to be an efficient way to represent the classification decision process [53], and is frequently applied to wetland and land cover mapping [54–66]. A rule-based system has obvious advantages and can be very effective if the classes are fixed and the knowledge in a particular domain can be encoded in a finite set of rules, where, for one rule at a time, all the data samples covered by that rule are removed from further search space. This approach can be used to decompose a multi-source, multi-sensor problem into a set of individual analyses and then combine their results in a separate analysis system that can perform a joint analysis [66–69]; however, this method has rarely been used in PFT classification.

Here, we design a new rule-based aquatic PFT classification method and a DT classification method by integrating time-series optical and radar data and DEM data (Figure 6). The steps for the rule-based aquatic PFT classification method are presented in Figure 6.

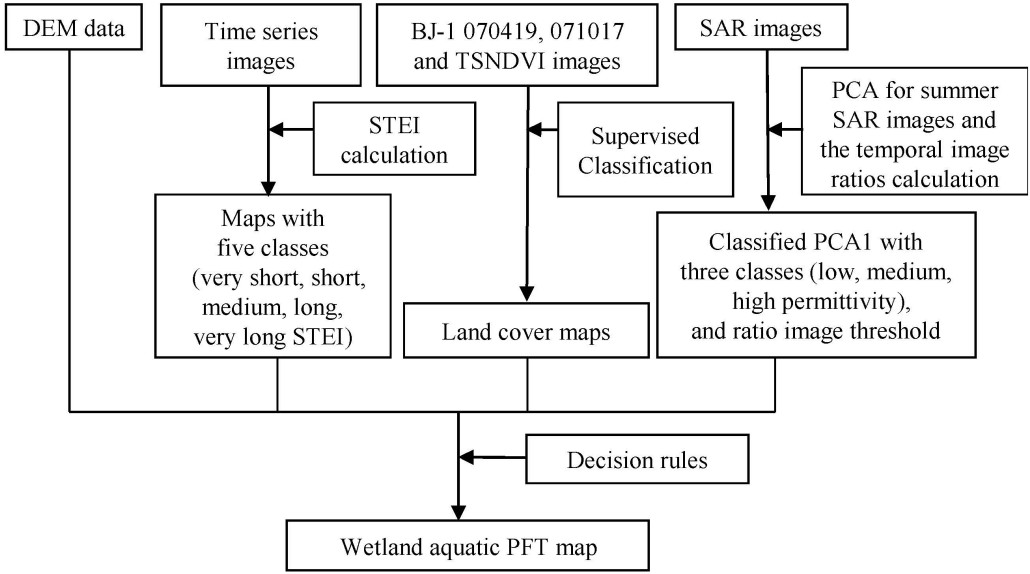

**Figure 6.** Flowchart of rule-based plant functional type (PFT) classification over Poyang Lake. STEI, submersion time estimation index; PCA, principal component analysis. Summer SAR images are ASAR images acquired on 29 June, 31 July, and 4 September 2007. Temporal image ratio refers to ratio between ASAR WSM images acquired on 10 June 2007 (WSM070610) and 29 June 2007 (WSM070629).

### 3.3.1. DEM Data

A digital elevation model (DEM) with approximately 92 m spatial resolution derived from the NASA Shuttle Radar Topography Mission (SRTM) [61,65,70] was used in this study. Although it cannot provide detailed information about the micro-topography of PFT distribution, it can provide vital information for determining the potential area of

wetlands [65,68,71]. Since the SRTM data have an absolute estimated height error of 11.25 m (1.6 σ) and a relative height error of 1.6–3.3 m [72], we used the slope data rather than the absolute elevation value, which is calculated using the ENVI topographic modeling package. Based on our field survey and a visual analysis of the natural wetland distribution boundary, we determined a slope less than 2° for the potential natural wetland area of the Poyang Lake region.

### 3.3.2. Submersion Time Estimation Index Calculation using Time-Series Images

Since there are not enough detailed and accurate bathymetry data for the lake bottom and the inundated marshland of the whole Poyang Lake region, we could not directly perform detailed aquatic macrophyte distribution mapping of the wetland. However, the water body submersion time, or the duration of inundation at each location, can reflect the dynamics of water depth and the relative elevation, since a location at lower elevation should have a longer submersion time [30]. The water level change can be determined more frequently if SAR data are combined with optical data for submersion time estimation.

We first extracted the water body from the negative NDVI images [38] of Beijing-1 data and the ASAR WSM images using density slicing and thresholding approaches, respectively. Then, we estimated the submersion time STEI($\omega$) as the average of monthly mean values of water bodies over the year:

$$STEI(\omega) = \frac{1}{12} \sum_{m=1}^{12} \left[ \frac{1}{n_m} \sum_{t=1}^{n_m} \omega_{m,t} \right], \tag{1}$$

where *STEI* ($\omega$) indicates the submersion time estimation index for the year, $m$ is the month, $n_m$ is the number of layers of water extent, and $\omega_{m,t}$ is the water extent layer extracted from Beijing-1 and ASAR WSM images for month m, with $t \in [1; n_m]$ and $\omega_{m,t} \in \{0,1\}$.

After calculating STEI, we further normalized it from 0 to 365, and classified the map into five classes: very short (<59.5 days), short (59.5–93.5 days), medium (93.5–158 days), long (158–223 days), and very long (>223 days). The thresholds were determined according to previous studies on the average submersion time of vegetation communities (Liu and Ye, 2000). These classes well reflected the relationship between freshwater aquatic PFT distribution and water level change characteristics at the regional scale.

### 3.3.3. Supervised Classification of Beijing-1 and Time-Series Normalized Difference Vegetation Index (TSNDVI) Images

Wang et al. [73] suggested that periods of water infilling and subsequent periods were the most representative for depicting seasonal floating and submerged aquatic macrophytes. Furthermore, the water infilling period is the best for depicting floating aquatic macrophytes, and the water subsiding period is the best for mapping submerged aquatic macrophytes. We used Beijing-1 images acquired on 19 April 2007 (BJ070419) and 17 October 2007 (BJ071017), representing water infilling and subsiding periods, respectively, to perform support vector machine (SVM) classifications. SVM has proved to be a superior method when the aim of the analysis is to map a set of specific classes or a single class with a limited number of training samples [74–77].

Training samples for PFT classification were selected using field samples acquired in April 2006 and March 2007 for the classification of image BJ070419, and field samples acquired in September 2006 for BJ071017. In order to better distinguish non-persistent emergent wetland PFTs, we performed SVM classification of TSNDVI images using the same training samples as those selected for the above-mentioned time-series data analysis. Training samples for non-PFT cover types were selected from each scene independently. Finally, we developed three land cover maps for the Poyang Lake region and performed accuracy assessment for each one using test samples from the field samples acquired in December 2007 to January 2008 and March 2008 for TSNDVI classification and April to May 2008, May 2009, and November 2008 for Beijing-1 classification.

### 3.3.4. Principal Component Analysis (PCA) and Temporal Ratio of ASAR WSM Time-Series Images

We found that the backscattering signatures for giant C4 grasses and emergent aquatic macrophytes in the summer of 2007 were higher than and distinctive from the other three PFTs due to their dominant double-bounced and volume backscattering. Previous research [35,67,78] has suggested that C-band SAR data are better adapted to detect standing water beneath the vegetative canopy and wet soil. Thus, we could use this information to discriminate emergent aquatic macrophytes and giant C4 grasses from the remaining PFTs. We first applied principal component analysis (PCA) to the summer ASAR WSM images acquired on 29 June, 31 July, and 4 September 2007. PCA was applied because it can summarize the variance of original images into fewer principal components. Here, we only chose the first principal component (PCA1) for subsequent analysis, because significant positive linear correlations were found between PCA1 and percent area flooded and soil moisture [62]. Then, supervised classification was conducted with the PCA1 result from each image to classify land cover types by using the interactively predefined training samples with low, medium, and high backscattering signatures [68,79]. Since the PCA analysis provided important information about vegetation canopy structural features, ground roughness, and soil or canopy moisture for different PFTs, the temporal ASAR images of 10 June and 29 June 2007 were suitable for distinguishing C3 (sedges and forbs) from other PFTs. Therefore, we calculated the ratio between them, and with the threshold as one of the rules (lowest class with ratio of 10 June and 29 June $\leq$ 0.85) to depict C3 (sedges and forbs).

### 3.3.5. Decision Rules for Final Classification

Knowledge-based decision rules were designed based on our field survey, expert experience, and the time-series analysis results. The conditions were established to combine information from the slope threshold for potential wetland area identification, STEI threshold for distribution pattern recognition of different PFTs with respect to water level changes, time-series Beijing-1 image classification for upland wetland area exclusion and aquatic bed wetland PFT differentiation, and PCA1 classification of time-series ASAR images for nonpersistent wetland PFT differentiation. If none of these were met or one pixel belonged to more than one class, the pixel was labeled as rapid succession, which occurs after flooding disturbance disrupts one and then another PFT. This is a perfect study area for understanding the inner-annual spatiotemporal changes of PFTs, which is beyond the scope of this research. The more rules that are included and the higher their precision, the more improved the classification results that can be achieved using this type of classification method. The decision rules developed here are based on knowledge we acquired from previous studies and experiments.

### 3.4. Decision Tree Method with Time-Series Optical, Time-Series Radar, and DEM Data

The decision tree method, as described below, was conducted based on the findings of the time-series analysis. The optimal thresholds were defined according to repeated experiments, in order to acquire the maximum consistent results with our field survey. The decision tree method is different from the rule-based method, because a decision tree algorithm directly constructs a global model without employing any local patterns during the construction process. If none of the above are met or one pixel belongs to more than one class, the pixel is labeled as rapid succession which occurs after flooding disturbance disrupts one of the PFTs and then another PFT. This is a perfect study place for understanding the inner-annual spatiotemporal changes of PFTs, which is beyond this research. The more rules that are included and the higher the precision of the rules, the more improved classification results that can be achieved in this type of classification method. The decision rules developed here are based on our knowledge acquired from previous studies and experiments.

## 4. Results and Discussion

### 4.1. Time-Series Data Analysis

The phenological and permittivity curves are shown in Figure 5. We found that floating aquatic macrophytes had relatively high NDVI values (Figure 5a) after nearly half a month growing in the ideal water level conditions, ranging from 10.7 m on 19 April 2007 to 13.7 m on 5 May 2007, then drawing back to 10.22 m on 25 May 2007 (Figure 7a). The backscattering coefficient for floating aquatic macrophytes had relatively high values, with an average of −20 dB, dominated by surface scattering on the ASAR WSM image acquired after the water retreated. Further discussion about PFT phenological variations can be found in Wang et al. [30].

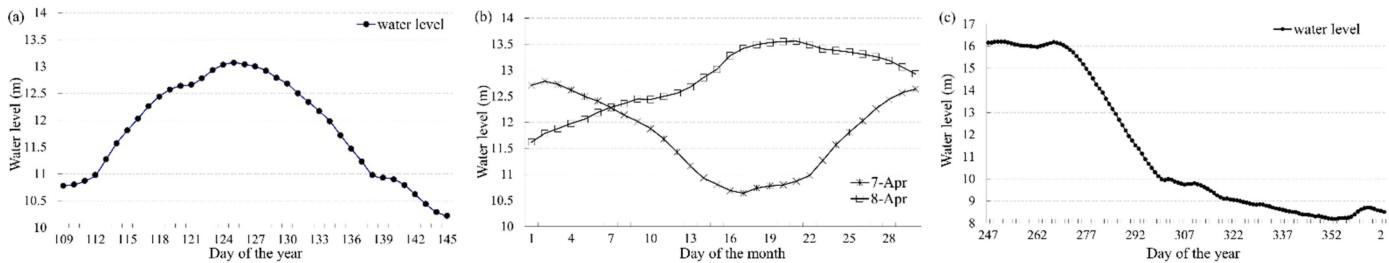

**Figure 7.** Temporal profile of water level data: (**a**) water level from 19 April to 25 May 2007; (**b**) comparison of water level in April 2007 and 2008; (**c**) water level from 4 September 2007 to 3 January 2008.

The backscattering coefficient for floating aquatic macrophytes had low values, with an average of −39 dB, dominated by specular scattering on the ASAR image acquired on 20 April 2008. The corresponding water level of 13.55 m was relatively high (Figure 7b), with most aquatic bed areas submerged compared to 19 April 2007, which is similar to locations supporting submerged aquatic macrophytes. For floating and submerged aquatic macrophytes, despite their high density and biomass during high-water-level periods, the backscattering coefficient was low due to dominance by specular scattering, making it difficult to distinguish these PFTs from open water. However, during water infilling and subsiding periods, it was possible to detect them from C-band ASAR WSM data.

Compared to floating and submerged aquatic macrophytes, the three other PFTs had high backscattering coefficients and biomass or density from April to early June. For emergent aquatic macrophytes and giant C4 grasses, the high soil dielectric constant reduces the transmission of radar waves, and the taller and denser vegetation morphology enhances the backscatter return, with an average of −17 dB. C3 (sedges and forbs) at higher elevation zones, despite their average height of only 60 cm above the soil, showed higher backscattering coefficients due to the dominance of volume scattering from high canopy coverage. On the other hand, C3 (sedges and forbs) at lower elevation zones with lower canopy coverage experienced disturbance by water level variation and were flooded during this period. Their horizontally oriented above-water stems allowed for double-bouncing of radar signals between the water surface and the stems, which considerably enhanced the backscattering coefficients.

Following the water level increase at the next growth stage after 10 June 2007, the submerged extent of C3 (sedges and forbs) gradually expanded, and the vegetation canopy became completely submerged by the end of July. During this time, there was little chance for radar signals to interact between canopy stems and water surface, and most of the radar energy was scattered away from the antenna direction, leading to a low backscattering coefficient of −42 dB on average for C3 (sedges and forbs) at the end of this period. At the same time, the NDVI value for C3 (sedges and forbs) decreased to the minimum value of the year, which also supports the findings above. By contrast, emergent aquatic macrophytes and giant C4 grasses were least disturbed by flooding and

still had a substantial amount of biomass above the water surface (Figure 5a). Therefore, they maintained high backscattering coefficient values of −20 dB on average during high-water-level periods because of strong volume scattering and double-bounce scattering. The backscattering signature of W0904 is similar to W0629 due to the similar water level (15.98 m on 29 June and 16.16 m on 4 September) and other environmental conditions at these two points in time.

During the period of subsiding water from 4 September to 28 October 2007 (Figure 7c), the water level decreased dramatically from 16.16 to 9.95 m, and the location with C3 (sedges and forbs) came out of the water with increased biomass or density. As a result, the backscattering coefficient values increased very quickly, from an average of −38 to −27 dB during this period. This is consistent with the trend of NDVI change for this PFT during this period. For floating and submerged aquatic macrophytes, their backscattering values did not show apparent variations around −37 dB during this period, because they were still dominated by specular scattering. The backscattering coefficients decreased from −15 to −22 dB and −17 to −29 dB from 4 September to 28 October 2007 for emergent aquatic macrophytes and giant C4 grasses, respectively, probably because of the reduced double-bounce backscattering due to the decreased water level. Following further water level decrease from 28 October 2007 to 3 January 2008, the backscattering coefficient increased to different degrees. The reasons for the variation during this period are complex.

Emergent aquatic macrophytes had the highest mean σ° values on 3 January 2008, which suggests that with declining biomass or density, this PFT could allow penetration of the C-band SAR signal to interact with the soil substrate. Their being 1.5 m higher than the surrounding area enhanced the surface roughness and SAR backscattering return. Floating aquatic macrophytes had the second highest mean σ° values on 3 January 2008 and increased quickly, ranging from −37 to −16 dB from 28 October 2007 to 3 January 2008. In part, this was because the water surface changed into a saturated mud flat with a higher dielectric constant. This reduced the radar wave transmission and enhanced the backscattering signal return. The withered plant shoots covering the ground increased the surface roughness. This also applies to the submerged aquatic macrophytes with a relatively high mean σ° on 3 January 2008. These results are not consistent with the results of Costa [80], who used Radarsat and JERS-1 images for mapping zonation of aquatic vegetation communities in the Amazon floodplain. His analysis indicated that at minimum water level periods, the backscattering values of C and L bands are the lowest. As the water level rises, so do the backscattering values. This may be because the predominant vegetation communities in Costa's study region were composed of large homogeneous stands of herbaceous semi-aquatic plants, pioneer shrubs, and various forest types, while ours is predominately grass-like aquatic communities. More detailed characteristics of Poyang Lake will be further explored in future work. There were no apparent backscattering variations for giant C4 grasses or C3 (sedges and forbs) during this period under relatively stable water level and moderate vegetation biomass and coverage conditions.

Numerous studies have investigated the classification of wetland vegetation in regularly water-influenced wetland, with large differences in data sources, classification algorithms and study periods. Zhang et al. [81] explored a hybrid approach on a single Landsat Thematic Mapper (TM) image for classifying coastal wetland vegetation classes. Several studies highlighted the importance of dual season and time-series remote sensing data for classification [51,80,82,83]. Most researchers use the same type of multi-source remote sensing data (either optical or radar) in order to increase the temporal resolution of data. However, while different PFTs may have similar features on optical data, they can be well distinguished on radar data, or the opposite. Our study thus contributes by taking the information of multi-seasonal optical and radar images for classification and proves that combining temporal backscattering signatures and NDVI of plant communities is the most effective.

*4.2. Rule-Based and Decision Tree Approach to Time-Series Optical and Radar Data*

The decision rules for final rule-based classification are described as follows:

- If a pixel belongs to TSNDVI class emergent or giant C4 grasses or C3 (sedges and forbs), and summer SAR PCA1 is high and STEI is very short, and slope <2°, then the pixel is classified as emergent.
- If a pixel belongs to TSNDVI class emergent or giant C4 grasses or C3 (sedges and forbs), and summer SAR PCA1 is medium and STEI is short, and slope <2°, then the pixel is classified as giant C4 grasses.
- If a pixel belongs to TSNDVI class emergent or giant C4 grasses or C3 (sedges and forbs) or floating, the ratio of ASAR 10 June and 29 June is lowest and STEI is medium, and slope <2°, then the pixel is classified as C3 (sedges and forbs).
- If a pixel belongs to BJ070419 class floating aquatic or farmland, and summer SAR PCA1 is low and STEI is long, and slope <2°, then the pixel is classified as floating aquatic macrophytes.
- If a pixel belongs to BJ071017 class submerged or BJ070419 class submerged, and summer SAR PCA1 is low and STEI is long/very long, and slope <2°, then the pixel is labeled as submerged aquatic macrophytes.

The rules for decision tree classification are described as follows:

- Slope <2°, ratio of ASAR 10 June and 29 June $\leqq$ 0.85, NDVI1130 $\geqq$ 0.2 for C3 (sedges and forbs)
- Slope <2°, PC1 of ASAR 29 June and 31 July and 04 Sep is from −12 to 3 for giant C4 and 3 to 30 for emergent
- Slope <2°, NDVI 19 April $\geqq$ 0.08 and NDVI 19 April $\leqq$ 0.41, C4 emergent decision result = 0, NDVI 30Nov $\leqq$ 0.3 for float
- Slope <2°, NDVI 17 October $\geqq$ −0.07, PC1 of ASAR 28 Oct, 18 December, 3 January, 7 February, 1 April, and 20 April $\leqq$ −25 discriminate aquatic bed and bottom from non-persistent emergent wetland and upland, PC2 of NDVI 19 April, 17 October, 30 November, and 01 January $\geqq$ −0.1, C4 emergent decision result = 0 for submerged

Figure 8a–d shows examples of Beijing-1 images acquired on 19 April and 17 October 2007, the first components of summer and winter ASAR WSM images, while Figure 8e–l shows maps of the five main PFTs from rule-based classification and the decision tree method.

We calculated the confusion matrices and kappa values using 305 field test samples (30 for giant C4 grasses, 20 for emergent aquatic macrophytes, 220 for C3 (sedges and forbs), 16 for floating aquatic macrophytes, and 19 for submerged aquatic macrophytes) and visually selected samples from the images with similar spectral and texture characteristics to our field samples. A comparison of producer's and user's accuracy (Figure 9) shows more accurate results with the rule-based method than the DT method. The decision tree method misclassified PFTs with higher commission and omission errors.

The producer's accuracy from rule-based classification increased by 3.1% for giant C4 grasses, 8.0% for emergent aquatic macrophytes, 3.5% for C3 (sedges and forbs), 8.4% for floating aquatic macrophytes, and 12.3% for submerged aquatic macrophytes (Figure 9a). The greatest improvement occurred in the class of submerged aquatic macrophytes, with increased accuracy from 85.4 to 97.7% (Figure 9a). This class tended to have a high omission error and be misclassified with C3 (sedges and forbs) by the DT method (Figure 8k,l). However, the user's accuracy for submerged aquatic macrophytes decreased slightly by 3.3% compared with the rule-based method (Figure 9b). The two lowest user's accuracy values among the five aquatic PFTs by the DT method were for emergent aquatic macrophytes (67.3%) and giant C4 grasses (76.1%). It is evident from the confusion matrix and visual comparison of spatial distribution (Figure 8e,f) that the DT method could not distinguish these two PFTs from upland farmland due to their short water submersion time, similar moisture status, and NDVI values, whereas the rule-based method, which conducted land cover mapping using temporal Beijing-1 images, overcome this problem and increased the user's accuracy by 25.6% for emergent aquatic macrophytes and 17.5% for giant C4 grasses.

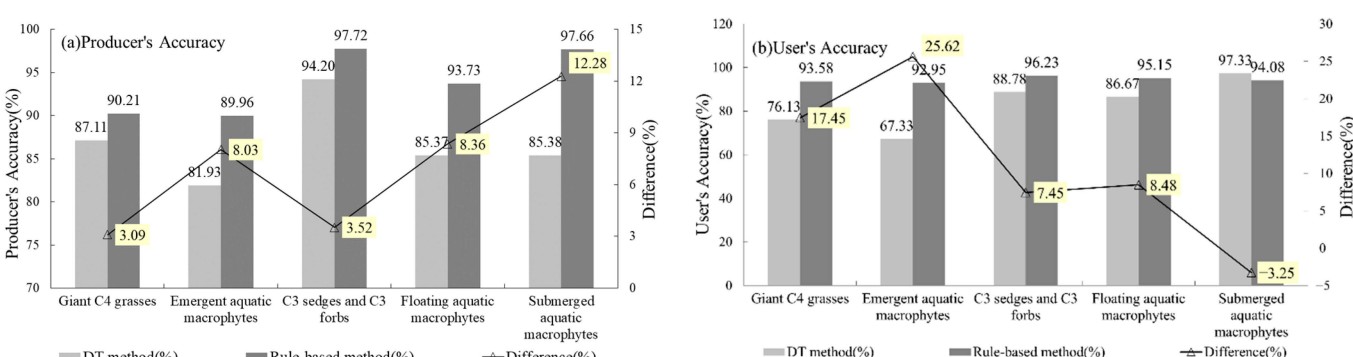

**Figure 8.** Beijing-1 images comprising first component of summer ASAR WSM PCA analysis and maps of five main PFTs from two classification methods. (**a**,**b**) Beijing-1 images acquired on 19 April and 17 October 2007, respectively; (**c**,**d**) PCA analysis of first component of summer and winter ASAR WSM image; (**e**,**f**) classification results of emergent aquatic macrophytes and giant C4 grasses by decision tree and rule-based methods; (**g**,**h**) classification results of C3 (sedges and forbs) by decision tree and rule-based methods; (**i**,**j**) classification results of floating aquatic macrophytes by decision tree and rule-based method; (**k**,**l**) classification results of submerged aquatic macrophyte by decision tree and rule-based methods.

**Figure 9.** Comparison of (**a**) user's and (**b**) producer's accuracy between decision tree (DT) and rule-based method.

*4.3. Benefits of Remote Sensing Analysis of Large Wetland Ecosystem with Strong Water Level Fluctuations and Future Work*

Remotely sensed data can provide useful information on the spectral characteristics and phenology of vegetation as well as landscape features that mediate the effects of flooding and other disturbances [30,39]. First, access to most wetlands is limited by water, mud, and management regulations. Considering the admittance, accessibility, and representativeness of different PFTs, field plots could be settled by means of high-resolution remote sensing data such as 5 m RapidEye multi-spectral images, 3 m PlanetScope multi-spectral images, or 0.5 m Skysat images. Although they do not cover large areas, they provide more detailed information of the season, time of the day, and phenological state of vegetation growth, which are difficult to achieve with field sampling. So high-frequency and high-resolution remote sensing data have an important role in the selection of priority target research sites and field sampling of PFTs in wetlands with strong water level fluctuations. Second, consistent with the results of Silva [84], our research results confirm that integrating ground-based ecological information with time series of remotely sensed data across multiple spatial scales, especially combining radar with visible-infrared sensors, proved to be very useful for mapping dominant PFTs and characterizing their biomass, canopy structure, and hydrological variations.

However, our results also raise several important questions for future discussion. Given the high seasonal variability of Poyang Lake flood extent and surface composition, our classification was intended to represent the first step in mapping "temporally" dominant PFTs from remotely sensed data in the context of the yearly cycle of wetland dynamics. This could be a useful baseline for subsequent analyses of inner-annual spatiotemporal variation in specific classes. Thus, an important research problem for the future is detecting inner-/inter-annual spatiotemporal changes of PFT, which could be used in simulation-based modeling of changes in wildlife habitats, plant biogeochemical cycling, and dynamics of hydrological conditions under different scenarios of water-level fluctuation, climate change, and human activities. In addition, these results have important implications for the analysis of other broad seasonal dynamics of aquatic ecosystems, such as plateau swamp, coastal, or saline wetlands. Further work should address extending this approach to these types of aquatic ecosystems where PFT habitat distribution is driven not only by water level dynamics, but also by temperature or salinity gradients.

**5. Conclusions**

By using Beijing-1 time-series NDVI and time-series ENVISAT ASAR images, we found that hydrophytes from different PFTs exhibit distinct phenological and structural characteristics due to the impact of the annual water inundation of Poyang Lake wetland. An analysis of backscattering properties of wetland PFTs indicated that non-persistent emergent wetland PFTs had high backscattering coefficients during the flooding period, which was due to high volume scattering and double-bounce backscattering. In the same period, aquatic bed PFTs had relatively lower backscattering coefficients due to their dominant specular scattering. During the period of low water level, their backscattering pattern varied. High backscattering values were observed during minimum water level periods, mainly because of the relatively high soil moisture even during the dry season and high ground surface roughness over non-persistent emergent wetlands, since the mass of plant residues covered the ground.

Phenological curves extracted from time-series Beijing-1 images and permittivity curves extracted from ASAR images can also distinguish different aquatic PFTs. By combining multi-sensor images, they can complement each other and produce more accurate PFT maps over the Poyang Lake region. The DT method has potential for mapping aquatic PFTs, whereas the rule-based method yielded more accurate results, with specific improvements as follows: (1) good distinction of non-persistent emergent wetland from upland farm land and (2) reduced omission errors for submerged aquatic macrophytes in DT-produced results. We suggest that the rule-based method is a better choice for PFT classification over

the Poyang Lake region. This method has proved to be an effective approach not only to distinguish different wetland types, as shown by Li and Chen [68], but also to differentiate various aquatic PFTs, as demonstrated in this research.

Plants are the primary producers of the Poyang Lake wetland ecosystem. Different PFTs have different ecological functions in this. Changes in distribution range, area, and species can directly reflect the evolution and health status of the wetland ecosystem. Therefore, accurate mapping and dynamic monitoring of Poyang Lake's PFTs are very important. Due to the large size and inaccessibility of the lake wetland, on-site detection is impractical, with limited human and material resources. Our method provides an effective solution for accurate mapping and continued monitoring of Poyang Lake PFTs. The approach and results here can provide data support and scientific guidance to help government agencies make management decisions for conserving and restoring the lake's wetland in the future, for example, through forecasting trends of plant functional patterns under extreme hydrological conditions to inform the development of countermeasures. Moreover, the method demonstrated in this paper could easily be extended to similar wetlands. With improvements in the spatial and temporal resolution of satellite remote sensing, phenological curves of different wetland vegetation types can be extracted more accurately to further improve classification accuracy. With the support of longer time-series data, this rule-based classification method can be used to study multi-year vegetation succession.

**Author Contributions:** Methodology, L.W.; software, F.D., K.C., and L.W.; validation, F.D. and L.W.; formal analysis, F.D., K.C., L.W. and I.D.; investigation, L.W. and I.D.; data curation, F.D. and K.C.; writing—original draft preparation, F.D. and L.W.; writing—review and editing, L.W. and I.D.; visualization, F.D.; funding acquisition, L.W. All authors have read and agreed to the published version of the manuscript.

**Funding:** This research was funded by the Scientific Institution Basal Research Fund, CAFS, China (2018HY-ZD0101); the Project of Yangtze Fisheries Resources and Environment Investigation from the MARA, P. R. China (CJDC-2017-24); Central Public-Interest Scientific Institution Basal Research Fund, CAFS, China (no.2020TD11).

**Data Availability Statement:** The data presented in this study are available on reasonable request from the corresponding author.

**Acknowledgments:** This work was supported by the National Natural Science Foundation of China (grant no. 30590370), and the National High Technology Research and Development Program of China (863 Program) (no. 2009AA12Z1462). We would like to thank Wan Wenhao from Nanchang University, Jiangxi Province, for his guidance during the PFT field investigations and his suggestion on PFT mapping; Beijing Twenty-First Century Science and Technology Development Co., Ltd., for supplying Beijing-1 satellite images; and the Dragon Programme for supplying ASAR WSM satellite images for this research.

**Conflicts of Interest:** The authors declare no conflict of interest.

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
