# Peer review of "Poyang Lake Wetland Classification Using Time-Series ENVISAT ASAR Data and Beijing-1 Imagery"

_water, doi:10.3390/w14203344_

Round 1
Reviewer 2 Report
Thank you very much for providing me with this opportunity to review this manuscript. Overall, I would like to congratulate the authors for this rigorous work. In the manuscript, all the statements, requiring citation, have been well referenced. The research rationale/hypothesis has been clearly communicated. The study objectives have been defined clearly. The introduction gives sufficient background information about the study. The study approach is consistent with the research objective. The methodology is appropriate and rigorous. The results appear complete and accurate and the findings are well‐supported by the data. The findings have been appropriately discussed in the context of previous literature.
However, the paper is poorly written. The English language for a scientific work is lacking. I would strongly suggest the authors to have this work proofread by some professional english language editing service.
Further, in addition to language issues, the study needs to highlight the limitations and future scope of the work regarding the wetland macrophyte classification, and how this method can be taken for a regional assessment.
Round 2
Reviewer 2 Report
Thank you for improving your manuscript.